# Diverging Grammaticalization Patterns across Spanish Varieties: The Case of *perdón* in Mexican and Peninsular Spanish

Marlies Jansegers [1,*] , Chantal Melis [2] and Jennie Elenor Arrington Báez [1]

[1]  Department of Linguistics, Ghent University, 9000 Ghent, Belgium; jennieelenor.arringtonbaez@ugent.be
[2]  Institute of Philological Research, The National Autonomous University of Mexico, Ciudad de México 04510, Mexico; cme@unam.mx
*  Correspondence: marlies.jansegers@ugent.be

**Abstract:** This study investigates the contemporary grammaticalized uses of *perdón* ('sorry') in two varieties of Spanish, namely Mexican and Peninsular Spanish. Methodologically, the investigation is based on a taxonomy of offenses, organized around the concept of 'face' and based on spoken data of Spanish from Mexico and Spain. This taxonomy turns out to be a fruitful methodological tool for the analysis of apologetic markers: it does not only offer usage-based evidence for previous theorizing concerning the grammaticalization process of apologetic markers, but also leads to a refinement of these previous results from a contrastive point of view. Evidence from both corpora suggests a more advanced stage in the grammaticalization process of *perdón* in Mexican Spanish, where it can be used not only as a self-face-saving device geared towards the positive face of the speaker, but also in turn-taking contexts oriented towards the negative face of the interlocutor. Peninsular Spanish, on the other hand, resorts to a more varied gamut of apologetic markers in these contexts.

**Keywords:** grammaticalization; apologetic markers; politeness; *face*; offenses; Mexican and Peninsular Spanish; *perdón*





## 1. Introduction

The speech act of apology can rightly be considered "one of the most profound human interactions" (Lazare 2004, p. 1). This is reflected in the omnipresence of apologetic markers in our everyday life: consider for example the number of daily messages we start with "sorry to bother you" or "sorry for my late reply", but also the vital importance of corporate apologies in customer service "sorry for the inconvenience" and of course, the recent avalanche of public apologies from governments, organizations and celebrities we see several times each week in the media. This pervasiveness of apologies has even led historians, philosophers, political and social scientists to coin the current era as an "Age of Apology" (cf., among others, Brooks (1999) and Gibney et al. (2007)).

From a linguistic point of view, these expressions such as (*I am*) *sorry* in English, *scusa* in Italian, *pardon* in French and *perdón* in Spanish have been qualified cross-linguistically in terms of grammaticalization, where the bleaching of the original semantic load gave rise to more procedural meanings (Molina 2011; Ghezzi and Molinelli 2019; Denoyelle 2020; Brenes Peña 2021). The contemporary Spanish marker *perdón*, for example, displays an ample gamut of uses that go beyond the original speech act of apology triggered by a previous offense and enter the realm of more procedural meanings related to various discourse-related functions. Consider the following examples:

(1) I: empezaban en los Andes// en los Alpes *perdón*/ los Andes están acá abajo/ en los Alpes// y entonces/ venían desde Francia/ caminando/ y está el camino de Santiago. (CSCM)[1]
'I: they started in the Andes/// in the Alps sorry/ the Andes are down here/ in the Alps// and then/ they came from France/ walking/ and there is the road to Santiago'.

(2) entonces yo digo <silencio/> *perdón* yo digo que este <vacilación/> cuando yo estaba por en <palabra_cortada/> entrar a la prepa era más fácil [...] (PRESEEA)
'then I say <silence/> sorry I say that this <hesitation/> when I was about to <unfinished word/> enter high school it was easier'.

(3) R: [(xx) eh: eh don Jesús *perdón* que lo interrumpa un segundito nada más tenemos a Abraham Mendoza en la línea estamos al aire para Panorama Informativo pues comentar específicamente lo que está sucediendo afuera y cómo se va a controlar (pues) esto parece ser incontrolable pero lo dejo al teléfono (CME)
R: '[(xx) eh: eh don Jesús sorry to interrupt you for a second, we have Abraham Mendoza on the line we are on the air for Panorama Informativo to comment specifically on what is happening outside and how it is going to be controlled (well) this seems to be uncontrollable but I'll leave you on the phone'.

(4) I: un rato llegué como a las/ dos y media de la mañana/ y me sacaron la bala hasta como por las nueve de la mañana
E: ¿como a qué hora? *perdón*
I: hasta las nueve (CSCM)
'I: for a while I arrived at about 2:30 in the morning/ and they took the bullet out of me until about 9:00 in the morning.
E: like at what time? sorry
I: only around nine o'clock'

(5) VV2F7 [v] Es que Johanson es la puta ama directamente VY2F8 [v] Esa mujer (())
VV2F7 [v]Luego está Chris Evans que para mí el capitán a -, a mí sí. VY2F8 [v]
*Perdón* pero no. Para mí el el más guapo el mejor lo que sea es Thor. Lo siento mucho. VV2F7 [v] Bueno también Thor no está mal no te voy a engañar (CORMA)
'VV2F7 [v] Is that Johanson is the fucking mistress directly VY2F8 [v] That woman (())) VV2F7 [v]Then there's Chris Evans who to me Captain a -, to me yes. VY2F8 [v] Sorry but no. For me the most handsome the best whatever is Thor. I'm sorry about that. VV2F7 [v] Well also Thor is not bad I'm not going to fool you'.

In example (1) *perdón* introduces a correction, whereas in (2) it is used to maintain the discursive thread. Moreover, *perdón* can also be used in contexts for turn-taking (3) or to request clarification (4) and even to attenuate upcoming criticism or difference of opinion (5).

Indeed, *perdón* has been related to multiple discourse-related functions. Fuentes Rodríguez (2009) includes *perdón* in her *Diccionario de conectores y operadores del español*, distinguishing between a "modal operator", used to attenuate the illocutionary force of speech acts threatening to the interlocutor (a request, a refusal, an interruption, etc.), and a "connective reformulation marker", appealed to in contexts of discourse self-repairs. Similarly, Brenes Peña (2021) organizes the procedural meanings of *perdón* developed by the original apologetic form along three dimensions: (1) metadiscursive, (2) interactional and (3) argumentative. She concludes that this leap from the sentence to the text as a unit of analysis of the lexeme *perdón* is a clear case of grammaticalization (or pragmaticalization, see below Section 2.2) where the bleaching of the original semantic load gave rise to more procedural meanings. From this perspective, then, *perdón* starts to compete with other markers displaying equivalent discourse-related functions such as *bueno* (disagreement, dispreferred responses and corrections), *oye/oiga* (to attract attention or act as a mitigator in controversial contexts), *o sea* (repair, utterance completion and clarification), *ahora* (disagreement marker), *este* (discourse flow, reformulation, hesitation), amongst many others.

However, if we identify *perdón* as a member of a paradigm that comprises other discourse markers (DM), the question arises as to what makes this marker unique compared with other (apparently) equivalent functional forms. The present study aspires to tackle this question and to discover the unique character of *perdón* by delving deeper into the grammaticalization process suffered by this apologetic marker. More precisely, some crucial questions remain unanswered:

- First, given the feature of persistence inherent to each grammaticalization process, what are the vestiges of the canonical model of apologies in the grammaticalized use of *perdón*?
- Second, if *perdón* is indeed the result of a grammaticalization process, to what extent can we relate its contemporary grammaticalized values to the original illocutionary meaning of the act of apology?
- Third, what are the underlying mechanisms and the subsequent compensatory forces for the bleaching of the original propositional illocutionary force of the original speech act of apology?

Taking into account the original semantics of *perdón* as well as the theoretical notions of *face* and *offense* crucial for the understanding of apologies, we propose an empirical study into the grammaticalization of contemporary *perdón* that aspires to both complement and deepen previous findings on this apologetic marker. Additionally, based on the idea that synchronic variation gives insight into ongoing change (Haverkate 1994; Wichmann et al. 2010; Winter-Froemel 2014; Lehmann [1995] 2015; Detges and Waltereit 2016; Gancedo Ruiz 2019) and the well-known fact that the use of politeness markers not only differs between languages, but also between varieties of the same language, we postulate the hypothesis that a comparison between two Spanish varieties might reveal diverging patterns of ongoing grammaticalization.

The outline of the paper is as follows: Section 2 provides a detailed theoretical description of the canonical model of apology (Section 2.1) and an overview of the previous studies on the origin and development of apologetic markers from a cross-linguistic point of view and the underlying mechanisms responsible for this semantic-pragmatic change (Section 2.2). Section 3 first discusses the method and data used in this study and then presents the results of a quantitative and qualitative analysis of the apologetic marker *perdón* in Mexican and Peninsular Spanish. Finally, in Section 4 these findings will lead to a detailed discussion on the contemporary uses of the marker, revealing diverging patterns of grammaticalization in both varieties.

## 2. Antecedents

### 2.1. The Canonical Model of Apologies

What all these contemporary apologetic formulas such as *(I am) sorry*, *scusa*, *pardon* and *perdón* have in common, is precisely their origin: they all developed from a speech act of apology. Therefore, in order to fully understand their contemporary behavior, it is crucial to first study the nature and essence of the canonical model of apologies.

Since Goffman (1971, pp. 138–48), the apology has been characterized as an essential element in a so-called *remedial interchange*. That is, the main purpose of an apology is to repair an offense committed by the speaker (the *offender*) against the interlocutor (the *offendee*) with the aim of restoring the social harmony or equilibrium between speakers (Edmondson 1981, p. 280; Leech 1983, p. 125). In the same vein, Holmes (1990, p. 159) provides the following definition of *apology*:

> An apology is a speech act addressed to B's face-needs and intended to remedy an offense for which A takes responsibility, and thus to restore equilibrium between A and B (where A is the apologizer, and B is the person offended).

This definition clearly highlights three important theoretical concepts linked to the study of apologies, namely (1) the speech act of apology itself, (2) the concept of *face* and—

more generally—linguistic politeness theory and (3) the apology as a reactive speech act that implies a previous offense for which the speaker takes (at least partial) responsibility.

First of all, according to the canonical model of apologies, an apology can be realized either explicitly or implicitly (cf., among others, Olshtain and Cohen 1983; Blum-Kulka et al. 1989; Holmes 1990; Trosborg 1995; Márquez-Reiter 2000; Harris et al. 2006; González-Cruz 2012). Instances of the latter, indirect, strategies are for example giving an explanation (*Parece que hoy no me concentro en nada* 'I just can't seem to concentrate today'), an offer of redress (*Le puedo comprar uno nuevo* 'I can buy you a new one') or a promise of forbearance (*No volverá a ocurrir* 'It won't happen again').[2] Alternatively, speakers can recur to explicit apologies which appear in the form of so-called illocutionary force indicating devices (IFIDs) as defined by Searle (1969, p. 62). For English, for example, the well-known taxonomy of apology IFIDs provided in the CCSARP coding manual (Cross Cultural Speech Act Realisation Patterns, cf. Blum-Kulka et al. 1989, p. 290) includes expressions with words such as *sorry*, *excuse*, *apologise* and *pardon*. Similarly, in Spanish, the IFIDs of apology can include both performative verbs such as *perdonar*, *disculpar*, *excusar*, *sentir*, *lamentar* (example 6) and formulas such as *lo siento* (*mucho*) y *perdón* (example 7):

(6) Oh, *discúlpame* amor, se me pasó el tiempo. (CREA, 1983)
'Oh, forgive me, honey, I lost track of time'.

(7) Oh, *perdón* -dijo el comisario-. Ignoraba que fuese usted viudo. (CREA, 1975)
'Oh, sorry, said the commissioner. I didn't know you were a widower'.

Second, since Brown and Levinson's (1987) seminal theory of politeness, apologies have been widely studied within the framework of politeness theory, reflecting as such the idea formulated by Holmes (1998, p. 217) that "the apology is quintessentially a politeness strategy". Crucial in their theory is the concept of *face*, first adopted by Goffman (1967, 1971). Building on Goffman's theory of face needs, Brown and Levinson (1987, p. 62) distinguish between *negative face* and *positive face*: the latter is defined as "the want of every member that his wants be desirable to at least some others", that is, the human need for recognition and affection (belonging). Negative face, on the other hand, is defined as "the want of every 'competent adult member' that his actions be unimpeded by others". This essentially refers to the human need to be autonomous. Other authors have (re)conceptualized negative face in Goffman's (1967, p. 5) terms of *territory* and *territories of the self*, alluding either to bodily, material, spatial, temporal or cognitive territory (e.g., Kerbrat-Orecchioni 1992; Bravo 1999; Bello 2015).

Ideally, then, every human being would be at the same time unanimously appreciated (positive face) but left alone (negative face) by others. However, this is an unattainable ideal as almost any interaction involves acts that are potentially threatening to one, or both, of these types of *face*. Such acts that infringe on someone's *face* have been labelled *Face Threatening Acts* (FTAs) by Brown and Levinson. In this view, then, the speech act of apology is considered to be face-saving for the hearer and face-threatening for the speaker (Olshtain 1989, p. 156). Indeed, as discussed above, an apology is typically aimed at face redress after committing an offense that has damaged the addressee's face. This is illustrated, for example in sentences such as (6) above and (8), where the speaker apologizes for not respecting and interfering with the temporal territory of the hearer:

(8) H2: Oye que *perdona* que hemos llegado tarde, ¿eh? pero es que... H4: [solapamiento de turnos] Ha sido culpa mía. (CORLEC)
' Hey, forgive us for being late, huh? but... H4: [overlapping of turns] It was my fault'.

However, as Deutschmann (2003, p. 43) points out, considering negative face redress as the only function of apologies would be a far too narrow view of this "versatile speech act". Indeed, as Holmes (1990, p. 162) already mentions, some apologies are geared towards positive face redress of the hearer. In the following example, the apology is supposed to redress an FTA to the hearer's positive face (in this case forgetting someone's name), explicitly attending to the hearer's wants and needs to be recognized:

(9)    E: muy bien Raquel / ¿tú sabes cocinar? - I: Rosa Rosa &lt;risas = "I"/ - E: ¡ay! *perdón* Rosa /
      ¿tú sabes cocinar? (PRESEEA)
      'E: very well Raquel / do you know how to cook? - I: Rosa Rosa &lt;laughs = "I"/ - E: oh!
      sorry Rosa / do you know how to cook?'

Although most apology studies have focused on redress of hearer's (positive or negative) face, it is well known that human beings not only try to protect the face of others, but also aspire to safeguard their own face (Goffman 1972). In this respect, Chen (2001) coined the term *self-politeness*, whereas other scholars prefer to talk about activities related to *self-image* preservation, referring to those situations where a speaker fears that his/her image might be endangered in the eyes of others by some utterance or potentially harmful action, i.e., an FTA to the positive face of the speaker (cf., among others, Bravo 2005; Hernández Flores 2005, 2008). Examples include apologies for social gaffes such as coughing and slips of the tongue.

In line with Deutschmann (2003, p. 43), the function of apologies as politeness markers should thus be considered as multi-*face*ted in the literal sense: although traditionally their use has been regarded as a way to redress (positive or negative) hearer's face, in many cases the use of apology markers is also aimed at maintaining or improving the speaker's face, or self-image. The theoretical concept of *face* is thus a very useful tool to account for the complex nature of apologizing, as long as multiple *faces* possibly involved in the interaction are taken into consideration. Therefore, in the present study, these four different axes will be considered: both the negative and positive face needs of both the hearer and the speaker.

A third key element in the study of apology markers is their reactive nature, and more precisely, the presence of an offense for which the speaker takes (at least partial) responsibility. Indeed, the offense or "object of regret" (Coulmas 1981, p. 75) is what essentially motivates an apology. As discussed above, within the theory of Brown and Levinson, offenses are seen as FTAs that can be geared towards both the negative or positive face of the speaker or hearer. Interestingly, although there seems to be little consensus among scholars on an operational taxonomy of offenses resulting in apologies, they do coincide in that the nature and severity of the offense will, to a great extent, determine the form of the subsequent apology (cf., among others, Holmes 1990; Kerbrat-Orecchioni 1992; Aijmer 1996; Wagner 1999; Deutschmann 2003). For example, bumping into someone accidentally will result in a different apology than would breaking someone's new phone. That is, there is reason to suggest that the offenses motivating an apology can be ordered along a continuum that proceeds from more serious to minor offenses.

However, when apologetic markers are studied in context, the empirical reality turns out to be even more complex than this. Indeed, several authors (Goffman 1971; Norrick 1978; Coulmas 1981; Ogiermann 2009; Ghezzi and Molinelli 2019) observe that the use of apologetic markers does not always constitute "genuine" speech acts of apology. Not uncommonly, explicit forms of apologies are uttered when the offense is minimal or even non-existent. In these cases, apologizing is rather a matter of routine, a mere formula used for the purpose of complying with social norms of good or polite behavior. Interestingly, in these formulaic apologies, speakers are more likely to resort to short, ritualized formulas such as *sorry* in English or *perdón* in Spanish. Compared with the more elaborated performative verbs such as *perdonar* ('to forgive') and *disculpar* ('to apologize'), these formulas are partially stripped of their original illocutionary force of apology and thus semantically bleached. This is why these expressions have also been characterized in terms of *grammaticalization*, which will be explained in the next section.

### 2.2. Grammaticalization of Apologies

The contemporary apologetic marker *perdón* seems to have originated as an ellipsis from the expression *te pido perdón* (Fuentes Rodríguez 2009, p. 251). This origin is what Spanish *perdón* has in common with its counterparts in other languages such as Italian *scusa* (*ti chiedo scusa > scusa*), French *pardon* (*Je vous demande pardon > pardon*) and English

*sorry* (*I am sorry* > *sorry*) (Molina 2011; Ghezzi and Molinelli 2019). Besides their formal resemblance, these expressions also seem to share their contextual origin. From the very beginning, apologies are linked to a religious context and from there they have spread to civil society through a complex process of secularization. In line with Kohnen (2017) and Williams (2018), Jucker (2019a) traces the long diachrony of apologies in the history of the English language. He convincingly demonstrates how the Christian act of penitence and repentance should be considered as a precursor of the modern apology, or at least as the first step in the semantic development of apologies. The same impact of religion has also been identified for French in the study of Denoyelle (2020). Medina López (2023) has very recently shown a similar diachronic development for Spanish apologies: expressions such as *perdón*, *me arrepiento*, *mi culpa*, *lo lamento* y *lo siento* originally are speech acts with considerable weight addressed to God that through a process of attenuation are gradually desacralized and eventually become used as speech acts that are often no more than "a token acknowledgement of some minor infraction" (Jucker 2019a, p. 17).

This formal and semantic development of apologies has been studied from a wide variety of theoretical perspectives on diachronic change and an ample gamut of specific diachronic processes have been identified to describe the rise of apologetic markers. Norrick (1979) for example, considers English *sorry* and *pardon* to be lexicalized pragmatic formulas. Similarly, for Aijmer (1996), (*I am*) *sorry* is a case of lexicalization, although recognizing "degrees of lexicalization on a scale of frozenness" (Aijmer 1996, p. 10). In view of Molina (2011), *sorry* is a case of pragmaticalization, linked to grammaticalization, and for Jucker (2019a), this diachronic process can be best characterized in terms of speech act attenuation. Within the Spanish realm, Brenes Peña (2021, p. 142) analyzes the uses of *perdón* deliberately as a case of "gramaticalización o pragmaticalización" without choosing between either term. Indeed, this is of course just a terminological question to describe the same underlying semantic-pragmatic process. As Brinton (2017, p. 34) points out, the choice between terms like *grammaticalization* and *pragmaticalization* seems to hinge not so much on the process itself but rather on what is encompassed by "grammar". The traditional conceptualization sees grammar as restricted to the morphosyntactic domain. Consequently, the process of grammaticalization focuses on the reduction of structure and form of linguistic units, while emphasizing the increase of their morphosyntactic dependence. If grammar is viewed from this rather narrow perspective, then pragmatic elements are excluded from the process of grammaticalization. Indeed, pragmatic markers may adopt textual functions, take care of discourse organization, express speaker stance or interpersonal values and can be used as politeness strategies, functions that "are not usually considered as the core business of grammar, if they are felt to be grammatical at all" (Van Bogaert 2011, pp. 315–16). However, different scholars have been arguing in favor of a more comprehensive, inclusive definition of *grammar* in order to go beyond the morphosyntactic level and also embrace discourse functions. As Traugott argues:

> I see grammar as structuring communicative as well as cognitive aspects of language. Grammar encompasses phonology, morphosyntax, and truth-functional semantics, and is rich enough to license interaction with the general cognitive abilities such as are involved in the speaker-addressee negotiation that gives rise to grammaticalization. These include information processing, discourse management, and other abilities central to the linguistic pragmatics of focusing, topicalization, deixis, and discourse coherence. (Traugott 2003, p. 626)

Such a broad conceptualization of grammar allows for pragmatic markers to be incorporated into the realm of 'grammar' and, thus, to be studied from the perspective of grammaticalization as well. Of course, such a comprehensive conception of grammar makes a term like pragmaticalization dispensable. As a consequence, the diachronic development of several discourse and pragmatic markers have been described in terms of grammaticalization (Castillo Lluch 2008; Wichmann et al. 2010; Molina 2011; Hancil 2018).

Indeed, at first sight, some of the traditional parameters linked to grammaticalization are reflected in the semantic-pragmatic evolution of the Spanish apology marker *perdón* (cf. Hopper 1991):

(i)   *Layering*: because the forms *perdón, perdona/e, lo siento, lo lamento* coexist in the same functional domain.

(ii)  *Divergence*: because the full expression *te pido perdón por*, the lexical noun *perdón* and the apologetic marker *perdón* exist side by side.

(iii) *Specialization*: according to previous studies (Fuentes Rodríguez 2009; Brenes Peña 2021) contemporary *perdón* specializes as a connective operating in three functional domains: (1) metadiscursive (2) interactional and (3) argumentative (see Introduction).

(iv)  *Persistence*: vestiges remain of the original (religious) meaning of *perdón* in the apologetic marker.

(v)   *Decategorialization*: loss of verbal/propositional characteristics.

However, the application of the parameters mentioned by Lehmann ([1995] 2015) is less straightforward: there is no obvious reduction of scope (condensation), no coalescence or morphological bonding, *perdón* does not become obligatory (obligatorification) nor paradigmaticalized (paradigmaticization) in so far as it does not join a grammatical paradigm. Instead of scope reduction (condensation), *perdón*—just like other pragmatic markers—rather exhibits scope extension and positional freedom.

Importantly, strictly speaking, there is even no semantic bleaching involved. As we already know from the diachronic studies mentioned above, the origin of contemporary *perdón* is not a word or a phrase with lexical or propositional content, but a speech act in itself with illocutionary force: *perdón* originates as a speech act in penitential acts and confessions to God and later becomes gradually desacralized to expressions of forgiveness and subsequently regret. Nowadays, it can even be used as a mere formulaic speech act, often without the presence of a clear offense. The ultimate goal has always been to maintain harmonious relations with God or the interlocutor. That is, in its origin, we have a speech act that is basically oriented towards the hearer, essentially interpersonally.

In other words, apologetic markers such as *perdón* in Spanish and *sorry* in English are not instances of a change from propositional content towards speech acts, but rather embody a process that affects the speech act itself, and more precisely, its illocutionary potential. They have an illocutionary function already at the onset and this illocutionary material is recruited for further illocutionary and discursive uses. In order to recognize this phenomenon in its own right, Arnovick (1999) and Claridge and Arnovick (2010, p. 187) coined this process *discursisation*. More recently, Jucker (2019a) uses the term *attenuation* for this process, i.e., a progressive weakening of its illocutionary force, and proposes a scale of speech act attenuation that is not unique to apologies but has wider applications. Applied to the history of apologies, he defines this as follows:

> The [...] history of apologies [...] shows a process of attenuation, that is to say their force becomes increasingly weaker, and at the same time the linguistic resources used to perform the speech act undergo a process of reduction and conventionalization. In its early form the speech act is semantically explicit and spells out its illocution while later it is reduced to a conventionalized expression that requires increased pragmatic processing for its interpretation. (Jucker 2019a, p. 7)

This "increased pragmatic processing" is what Claridge and Arnovick (2010) call "pragmatic strengthening".

Based on this theoretical background, in the remainder of this paper, we will present the results of an empirical, comparative study of the uses of *perdón* in contemporary Spanish. By comparing two different varieties of Spanish (Mexican and Peninsular Spanish) we aspire to add new insights into this ongoing debate on the grammaticalization, discursisation or attenuation of apology markers.

## 3. Case Study: *Perdón* in Mexican and Peninsular Spanish

### 3.1. Methodology and Corpus

The data used to investigate the contemporary uses of the apology marker *perdón* come from a sizeable, manually annotated corpus consisting of spoken spontaneous conversations and interviews, recorded during the last quarter of the 20th century and the first decades of the 21st century. The following list of existing, spoken corpora were exploited: Corpus del Proyecto para el Estudio Sociolingüístico del Español de España y de América (PRESEEA), América y España español coloquial (AMERESCO), Corpus Sociolingüístico de la Ciudad de México (CSCM), Corpus del Habla de Baja California (CHBC) and Corpus Michoacano del Español (CME) for Mexican Spanish and Corpus Oral de Madrid (CORMA), Valencia Español Coloquial (Val.Es.Co.), Corpus Oral de Referencia del Español Contemporáneo (CORLEC), Corpus integrado de referencia en lenguas romances (C-ORAL-ROM) and Corpus del Proyecto para el Estudio Sociolingüístico del Español de España y de América (PRESEEA) for Peninsular Spanish. In a first phase we collected all occurrences of *perdón* together with its near-synonymous apologetic markers such as *lo siento* and forms derived from the performative verbs *perdonar* and *disculpar*. This first step allowed us to quantitatively assess the overall frequency and productivity of *perdón* compared with its near-synonymous expressions. Following this objective, we retrieved a total of 769 occurrences: 363 cases for Mexican Spanish and 406 for Peninsular Spanish. From this corpus, we then selected only the instances of *perdón*, yielding a total of 500 instances (299 for Mexico and 201 for Spain) that were subjected to a fine-grained qualitative analysis.

For this qualitative study, we analyzed the speech act of apology from the perspective of the offenses that motivate the apology and built on the hypothesis presented in the theoretical introduction (see Section 2.1) that the nature and severity of the offense partly determine the forms chosen to apologize. The analytical tool used is a taxonomy of offenses, organized around the concept of face as conceptualized by Goffman (1967) and Brown and Levinson (1987). As Deutschmann (2003, p. 62) already pointed out, unfortunately there has been little consensus over the taxonomy of offenses in past studies. Consequently, we decided to propose our own taxonomy of offenses in Spanish. This taxonomy is based on the empirical data from our own corpus and supported by a meticulous review of the literature on the subject. Table 1 below summarizes our taxonomy of offenses motivating apologies in Spanish, organized around the concept of face for both the speaker and the interlocutor:

**Table 1.** Taxonomy of offenses in Spanish, organized in terms of face.

| Negative Face (Hearer) | Positive Face (Hearer) | Positive Face (Speaker) |
|---|---|---|
| Invasion of spatial territory | Criticism or disagreement | Slips of the tongue (*lapsus linguae*) |
| Interference with temporal territory | Lack of consideration | Censored language |
| Interruptions | Rejection of offers or invitations | Inappropriate behavior |
| Damage to belongings | Breach of promise | Social gaffes |
| Violation of the right not to be distracted | | |
| Obligation to do something | | |
| Obstruction of plans | | |

As Table 1 shows, this taxonomy embodies an attempt to relate, in a precise manner, each type of offense to the typical traits associated with the positive and negative face of both interlocutors. As such, it encompasses three broad classes, made up of several subclasses: (1) offenses that damage the hearer's negative face (2) offenses aimed at the hearer's positive face and (3) offenses that threaten the speaker's positive face.[3] Applied to our corpus data, this means that all 769 occurrences were analyzed manually, identifying for each of them, (a) the specific formula of apology (*perdón*, *perdona/e*, *disculpa/e*, *lo siento*), (b) the type of offense, (c) the face affected (positive face/negative face), and (d) the orientation of the face (speaker/hearer). In the following sections, we will first present

the overall quantitative results (Section 3.2) and then turn to the more in-depth qualitative analysis of the marker *perdón* (Sections 3.3 and 4).

### 3.2. Quantitative Results

First of all, the question arises as to what are the most commonly used explicit apology formulas in Spanish. Table 2 displays the overall frequency of each form in the Mexican and Peninsular corpus.

**Table 2.** Apology forms in Mexican and Peninsular Spanish. ($\chi^2$ = 192; df = 3, *p* < 0.001).

| Forms | Mexico | | Spain | |
|---|---|---|---|---|
| | # | % | # | % |
| *disculpar* | 49 | 14% | 12 | 3% |
| *perdonar* | 11 | 3% | 158 | 39% |
| *lo siento* | 4 | 1% | 35 | 9% |
| *perdón* | 299 | 82% | 201 | 49% |
| Total | 363 | 100% | 406 | 100% |

It is striking that, although all apology forms are present in both varieties, their distribution diverges considerably between Mexican and Peninsular Spanish. What both varieties have in common—and what immediately catches the eye, of course—is the clear supremacy of *perdón* in this pragmatic domain of Spanish apologetic markers. However, this dominance is much more evident in Mexican Spanish, where *perdón* covers 82% of all cases and outperforms by far all other forms in the apologetic domain. In Peninsular Spanish, on the other hand, there still seems to be a stronger competition between *perdón* (49%) and the performative verb *perdonar* (39%). These quantitative differences between both varieties possibly point towards some important underlying qualitative divergences related to the functional scope of the marker *perdón* in Mexican and Peninsular Spanish. Indeed, in light of Zipf's (1949) *Principle of Economic Versatility* according to which frequency of use is correlated with semantic versatility, the higher frequency of *perdón* within the functional domain of apologetic markers might correlate with some semantic-pragmatic enrichment, eventually leading to a grammaticalized discursive formula. Additionally, the frequency differences between both varieties also suggest some dialectal variation in this grammaticalization process. In order to interpret these quantitative data, in the next section we turn to an in-depth contrastive analysis of the most frequent apology marker in both varieties, *perdón*.

### 3.3. Perdón in Méxican vs. Peninsular Spanish

In order to define the underlying mechanisms responsible for the frequency differences in both varieties, we analyzed each occurrence of the corpus for three qualitative variables, namely (a) the specific type of offense, (b) the face affected (positive/negative) and (c) the orientation of the face (speaker/hearer). Considering these variables, the distribution for the specific forms in the Mexican corpus is represented in Table 3:

**Table 3.** Relation between (orientation of) face and form of apology in Mexican Spanish.

| | *Disculpar* | | *Perdonar* | | *Lo Siento* | | *Perdón* | | Total | |
|---|---|---|---|---|---|---|---|---|---|---|
| | # | % | # | % | # | % | # | % | # | % |
| Positive face S | 15 | 7% | 3 | 1 | - | - | 196 | 92% | 214 | 100% |
| Negative face H | 22 | 18% | 6 | 5% | 1 | 1% | 90 | 76% | 119 | 100% |
| Positive face H | 12 | 40% | 2 | 7% | 3 | 10% | 13 | 43% | 30 | 100% |

The frequencies in this table reveal a very interesting profile of the marker *perdón* in the Mexican corpus. Compared with the other possible apologetic forms at the speaker's

disposal, *perdón* turns out to be the default form for the protection of the speaker's positive face (92%), highly preferred after offenses geared towards the hearer's negative face (76%) and much less used for offenses towards the hearer's positive face (43%). As we saw in the taxonomy of offenses presented in Table 1, the first category of offenses towards the speaker's positive face encompasses slips of the tongue (example 10), censored language (example 11), inappropriate behavior (for example leaving a place abruptly, see example 12) or social gaffes such as sneezing or burping (example 13):

(10)   […] o sea/ prácticamente toda la secundaria/ toda la prepa *perdón*/ vivía en su casa […] (CHBC)
'I mean/ practically all of secondary school/ all of high school excuse me/ I lived in his house […]'

(11)   B: [sí otra] vez no/ no mamá cuando una gente es *perdón* pero <énfasis t="pronunciación_marcada">imbécil</énfasis>/ es imbé[cil toda su vida] (AMERESCO)
'B: [yes again] no/ no mom when someone is sorry but <emphasis t="pronunciation_marked">imbecile</emphasis>/ is imbe[cil all his life]'

(12)   A: y yo estoy bien así o sea<alargamiento/>// con el respeto se basa porque <fsr t="pus">pues</fsr>// luego tardo ocho días en verla// diez días quince días dependiendo/// (1.6) <fsr t="horita">ahorita</fsr> la vi ¿cuándo?/ ¿martes?/// (2.2) tal vez no la hubiera visto si hubiera trabajado no la hubiera visto/ hasta// el miércoles o el jueves/// (1.8) es así/// (1.2) nos mantenemos así respetados/ pero/// (1.8) <fsr t="horita">ahorita</fsr>se va a enojar si no voy <risas/>// porque// es que<alargamiento/>/// (1) me retiro señora/ *perdón*
C: ¿[sabe que estás] conmigo?
B: [<ininteligible/>]
A: le dije [que fui a mi]sa (AMERESCO)
'A: and I'm fine like this, that is to say</>/// with respect it is based because <fsr t="pus">then</fsr>/// it takes me eight days to see her/// ten days fifteen days depending//// (1.6) <fsr t="horita">ahorita</fsr> I saw her when?/// Tuesday?//// (2.2) maybe I wouldn't have seen her if I had worked I wouldn't have seen her//// until Wednesday or Thursday//// (1.8) it's like this/// (1.2) we keep each other respected that way//// but/// (1.8) <fsr t="horita">ahorita</fsr> she'll get mad if I don't go <laughs/>/// because/// it's that<lengthening/>/// (1) I'm leaving madam/ sorry.
C: [does he know that you are] with me?
B: [<unintelligible/>]
A: I told him that I went to Mass'

(13)   I: fue muy / a mí me gustó ser cúrpite esa vez / y para las danzas me gustó mucho ser maringuilla // y fíjate que hasta / para cuestión // <ruido = "eructo"/> *perdón* / de vestuario // (PRESEEA)
'it was very / I liked being a *cúrpite* that time / and for the dances I really liked being a *maringuilla* // and notice that even / for the question // <noise = "burp"/> sorry / of costumes'

As can be seen from the examples above, this category encompasses situations where the offense for the addressee is minimal, even non-existent. In these cases of "formulaic" apologies (Deutschmann 2003, p. 46), the presence of an IFID of apology seems to be rather a matter of routine, primarily motivated by the desire to "evince good manners" (Norrick 1978).

Besides the offenses towards speakers' positive face, *perdón* is also very frequent in contexts with offenses towards the hearer's negative face. In this category, *perdón* turns out to be extremely productive in the case of interruptions and turn-taking contexts (62 out of 90 occurrences, see example 14):

(14)   I: un rato llegué como a las/ dos y media de la mañana/ y me sacaron la bala hasta como
       por las nueve de la mañana
       E: ¿como a qué hora? *Perdón*
       I: hasta las nueve (CSCM)
       'I: for a while I arrived at about 2:30 in the morning/ and they took the bullet out of me
       until about 9:00 in the morning.
       I: about what time? Sorry
       I: until nine o'clock'

These examples of turn-taking contexts illustrate that *perdón* is indeed moving towards discourse organization, which coincides with the interactional dimension distinguished by Brenes Peña (2021) in the grammaticalization process suffered by *perdón*.

Finally, compared with these two categories, the offenses oriented towards the hearer's positive face seem to somewhat resist the use of *perdón*. In these cases, *perdón* starts to compete with the more elaborate, performative verb *disculpar* (resp. 43% vs. 40%). This is the case, for example, with disagreements and criticisms, as illustrated in example (15):

(15)   eh nosotros ya ya veíamos que que la gente pues se limitaba exclusivamente a eso el
       turismo en sí no: no prolifera:ba no era pero le le vuelvo a insistir le vuelvo a insistir esto no
       *perdón* no lo veo yo como: que sea algo en su totalidad de de echado a perde:r de: de que
       quisiéramos que ya no vinieran no no no no este esto es lo que no- nos da vida porque
       mucha gente se da cuenta de lo que realmente es- [CME]
       'eh we already saw that people were limited exclusively to that, tourism itself was not
       proliferating but I insist again I insist again this is not sorry I do not see it as something
       totally spoiled that we would not want them to come anymore no no no no hum this is
       what does not-it gives us life because many people realize what it really is'

Interestingly, as discussed above (see Section 2.1), some previous studies have suggested that the offenses motivating an apology can be ordered along a continuum that proceeds from more serious to minor offenses. Applying this rationale to our tripartite taxonomy of offenses, we can indeed arrange the three categories described above along this continuum. At the pole of serious offenses are threats to the positive face of the hearer, such as criticism or disagreement. These instances can be classified as more serious offenses on the grounds that they challenge the other's social dignity (his/her positive face-wants of belonging). Next in line are the offenses threatening the hearer's negative face, as they involve violations of one's claims to privacy and freedom from impositions and impediments. Compared with the previous category of serious offenses, these are minor offenses because in these circumstances the expression of apology does not express true regret on the part of the speaker but serves instead as a "token acknowledgment of some minor infraction" (Jucker 2019a, p. 17), "associated with the implied message 'please, don't think I'm rude' as a socially coded meaning" (Williams 2018, p. 159). Finally, in our third category of offenses we have subsumed perturbations in the flow of discourse (vacillations, reformulations, etc.), the use of improper language, the so-called social gaffes (burping, sneezing, etc.) and other incidents of this kind. These cases are likely to produce shame or embarrassment in the speaker himself/herself, but do not imply any offense at all to the interlocutor. Therefore, they occupy the other pole of the continuum.

Interpreting our Mexican data in light of this continuum, we can conclude that Mexican Spanish seems to generalize *perdón* primarily as a self-face-saving device (positive face of speaker), essentially geared towards the face-wants of the speaker but without real offense towards the interlocutor (92%). To a lesser degree, it is also used for minor offenses towards the hearer's negative face (76%), but for the more serious offenses, it competes with the performative verb *disculpar* (43%).

Table 4 compares these tendencies with the Peninsular Spanish corpus data:

**Table 4.** Relation between (orientation of) *face* and form of apology in Peninsular Spanish.

| *Face* Affected/Orientation | *Disculpar* | | *Perdonar* | | *Lo Siento* | | *Perdón* | | Total | |
|---|---|---|---|---|---|---|---|---|---|---|
| | # | % | # | % | # | % | # | % | # | % |
| Positive face S | 4 | 2% | 31 | 19% | 6 | 4% | 125 | 75% | 166 | 100% |
| Negative face H | 7 | 4% | 88 | 50% | 15 | 9% | 66 | 37% | 176 | 100% |
| Positive face H | 1 | 1% | 39 | 61% | 14 | 22% | 10 | 16% | 64 | 100% |

The overall distribution of *perdón* stands out: it seems to run along the continuum of seriousness in the same direction as in Mexico, but with some striking diatopic differences. Although both Spanish and Mexican data favor the use of the form *perdón* for offenses towards the positive face of the speaker, Peninsular Spanish does not reach the same level of productivity as Mexican Spanish (75% in Peninsular vs. 92% in Mexican Spanish). Contrary to what happens in the Mexican corpus, in the context of minor offenses towards the negative face of the hearer, Peninsular *perdón* loses its privileged status and competes with its performative counterpart *perdonar* (resp. 37% and 50%). Finally, for the more serious offenses towards the positive face of the interlocutor, *perdón* is very rare (16%). In these cases, the use of *perdonar* also prevails (61%) and—to a lesser extent—*lo siento* is used (22%). In the next section, we will interpret these results against the broader theoretical background of grammaticalization theory and semantic-pragmatic change.

## 4. Discussion: Diverging Patterns of Grammaticalization

At first glance, our corpus-based study on *perdón* in Mexican and Peninsular Spanish seems to confirm the findings of previous studies in that the wordform *perdón* has undergone a grammaticalization process resulting in multiple discourse-related functions. Indeed, Fuentes Rodríguez (2009) includes *perdón* in her *Diccionario de conectores y operadores del español*, distinguishing between a "modal operator", used to attenuate the illocutionary force of speech acts threatening to the interlocutor (a request, a refusal, an interruption, etc.), and a "connective reformulation marker", appealed to in contexts of discourse self-repairs. Similarly, Brenes Peña (2021) organizes the procedural meanings developed by the original apologetic form along three dimensions: metadiscursive (text connective in reformulations, hesitations, repetitions, etc.), interactional (turn-taking system, attention-getter in discourse openings, introducing requests of repetitions and explanations), and argumentative (disagreement marker). Analogous views have been expressed in relation to English *sorry* (see Section 2 above). From the perspective of these authors, the grammaticalization process undergone by *perdón/sorry* is comparable to that of other DMs and developed into a wide array of discourse-related functions that compete with other DMs displaying equivalent functions such as *bueno* (disagreement, dispreferred responses and corrections), *oye/oiga* (to attract attention, or to act as a mitigator in controversial contexts), *o sea* (repair, utterance completion and clarification), *ahora* (disagreement marker), *este* (discourse flow, reformulation, hesitation), amongst others.

However, this competition of different DMs for one and the same function leaves the motivation behind the choices speakers make between available forms unexplained. If we identify *perdón* as a member of a paradigm that comprises other items like *bueno*, *o sea* or *este*, the question arises as to what extent these DMs are interchangeable and what motivates speakers to choose one DM over another to express the "same" function. More specifically, the present study focused on *perdón* in order to answer the question of what makes this marker unique compared with other (apparently) equivalent functional forms. By taking into account the type and seriousness of the offense and the face affected, the present study aspired, then, to both complement and deepen previous findings on this apologetic marker.

Under our proposal, contemporary *perdón* is a polysemous item (see Fischer 2006), basically associated with three pragmatic meanings. The interpretation of the formula

depends on the concrete discourse contexts in which it is used and interacts in a critical way with the type of offense:

(i) *Perdón₁* is a genuine expression of regret that is essentially hearer-supportive, but it comes along with a secondary and inferable message of interest to the speaker ('I know the norm; I usually do not offend people');

(ii) *Perdón₂* relates to minor social infringements; the illocutionary component of regret is diluted, while the self-protective inferential message gains in prominence;

(iii) *Perdón₃* is a grammaticalized self-face-saving device; the remedial move exclusively targets the face-wants of the seemingly apologizing individual. This category also includes the cases related to the flow of discourse.

### *Perdón₁: expression of regret geared towards the hearer*

As explained above (see Section 2), the form *perdón* originates as an ellipsis of the performative verbal expression *pido perdón*, and, up to the present, it can be used as an IFID for expressing genuine apologies. This origin already sets apart *perdón* from other DMs in the same paradigm: it is important to bear in mind that—contrary to what usually happens to other DMs—the point of departure for *perdón* rests in an element that already has an illocutionary force at the onset and this illocutionary material is recruited for further illocutionary and discursive uses (Claridge and Arnovick 2010). Indeed, from a canonical point of view, a speech act of apology is prompted by a wrongdoing—an offense—committed by one person against another, for which the offender takes at least partial responsibility and apologizes to the offended individual in an attempt to repair the damage inflicted on their relationship. In this scenario, the illocutionary meaning of the utterance of apology is an expression of regret (Norrick 1978) on the part of the speaker, who communicates his/her emotional state regarding the situation, and the desired perlocutionary effect of the apology has to do with the hope of being forgiven by the addressee (Norrick 1978; Edmondson 1981). This also explains the common view that apologies are fundamentally hearer-supportive and can therefore be defined as a manifestation of "polite" behavior targeted at the addressee's face-needs, i.e., attentive to the other's concern for his/her social image, and intent on boosting the other's sense of self-worth potentially harmed by the offense (e.g., Owen 1983; Trosborg 1987; Olshtain 1989; Blum-Kulka et al. 1989; Wagner 2004; González-Cruz 2012). In other words, from a canonical point of view, apologies are by nature intersubjective, if this notion is understood as the "expression of Speaker attention to the 'self' of addressee [...] in the social sense of paying attention to their 'face' or 'image needs' associated with social stance and identity" (Traugott 2003, p. 128). Normally, (inter)subjective phenomena arise when lexical items develop pragmatic functions, but expressive speech acts like apologies are interactional by nature and convey attitudes and evaluations of the involved parties as part of their essence.

It is worth noting that apologizers themselves also obtain some benefit as a product of their willingness to engage in repair work. This is suggested in various studies (Holmes 1990; Haverkate 1994; Ogiermann 2006, 2015), and is elaborated upon in Meier (1995, pp. 388–89):

> Note that in stark contrast to B/L and those who incorporate their framework, I posit Repair Work to be an image-saving device as regards the Speaker (not the Hearer), making S's image the central figure. Concern for H's face is only a by-product of the attempt to save S's face, "'an altruism in egoism" as so aptly puts it. Repair Work is thus an attempt to show that the Speaker is a 'good guy' (despite having violated a social norm) and can be relied upon in the future to act predictably in accordance with the social norms of a particular reference group (i.e., to act appropriately). This is a type of reaffirmation of shared values, an uncertainty reduction, which helps to assure S's membership in the group wherein she or he can derive the same benefits from co-members' predictable behavior as they can from S's.

Although we may disagree with Meier's exclusive emphasis on the speaker-oriented dimension of genuine apologies, the idea that speakers simultaneously pursue subjective goals when they apologize makes sense, considering Goffman's (1972) hypothesis about both other-directed and self-directed facework moves that occur in social interactions.

As a way of integrating this idea in our definition of genuine (serious) apologies, we propose, drawing on Boye (2023), that an additional, underlying meaning akin to 'I know the norms of appropriate behavior prevailing in our community and I usually act accordingly' exists as a pragmatic inference with "discursively secondary status", which has the potential of becoming conventionalized or grammaticalized in contexts where no real speech act of apology is performed.

This use of *perdón*$_1$, related to more serious offenses, appears in contexts of disagreements and criticisms (see example 15 above) and also in case of lack of consideration, where the speaker offends the hearer by ignoring, for example, his name (16):

(16)   BAR3M1 [v] Ahora, la tostada. Gloria, solo mermelada sin mantequilla ¿no? CBAR3F6 [v] No no, es Paula. BAR3M1 [v] Ah, Paula *perdón*, me confundio' de nombre. (CORMA) 'BAR3M1 [v] Now, the toast. Gloria, just jam with no butter, right? CBAR3F6 [v] No no, it's Paula. BAR3M1 [v] Ah, Paula *sorry*, I got the wrong name.'

### *Perdón*$_2$: minor social infringements

Moving further along the continuum of seriousness, consider now the intermediate cases in which people apologize for what seems to be rather minor offenses, primarily motivated by the desire to evince good manners (Norrick 1978). The impression is that, in these circumstances, the words of apology do not express true regret on the part of the speaker—meaning that their original illocutionary force is weakened—serving instead as a "token acknowledgment of some minor infraction or mishap" (Jucker 2019b, p. 20), "associated with the implied message 'please, don´t think I'm rude' as a socially coded meaning" (Williams 2018, p. 159). Thus, in comparison with genuine apologies, as defined above and drawing on Boye (2023), we can speak of a shift in the relative prominence of the expression of regret, signaled by *perdón*, and the inferable device of self-protection. What in the case of genuine apologies only occupies "discursively secondary status"—i.e., the speaker's concern with projecting the image of a person who complies with the rules of socially sanctioned behavior—now comes to occupy center stage, while the remedial work undertaken for the hearer's sake recedes into the background.

Examples of this category are interruptions in turn-taking contexts and different kinds of impositions on the hearer, as seen above in (14) and again in the following excerpt (17):

(17)   oye pero pero / ¿cómo? / bueno / *perdón* que te regrese al tema / [. . .] E: eso ya me interesa particularmente / porque creo que me debe interesar // este / pero cómo es / o sea / entonces / ¿el virus del papiloma / ya es cáncer? (PRESEEA) 'hey but / how? / well / *sorry* to get back to the topic / [. . .] E: I'm particularly interested in this / because I think I should be interested // in this / but how is it / I mean / so / is the papillomavirus / already cancer?'

It is clear that in this kind of examples, the offense rather relates to a minor social infringement, while the self-protective inferential message gains central stage.

### *Perdón*$_3$: grammaticalized self-face-saving device

The third type of situation in which *perdón* occurs embraces all those cases where one is at a loss trying to determine the "offense" inflicted on the addressee. As mentioned in the previous section, these situations correspond to perturbations in the flow of discourse (vacillations, reformulations, etc.), the use of improper language, the so-called social gaffes (burping, sneezing, etc.) and other incidents of this kind, which are likely to produce a sense of shame or embarrassment in the speaker himself/herself. Various uses of *perdón*$_3$ were shown in (10) to (13) above. Another case of *lapsus linguae* is illustrated in (18):

(18)    cuando estaba ya, este, trabajando en San Juanito San Juanico, *perdón* dónde varias salchichas de gas estallaron. (CHBC)
‘when I was already, huh, working in San Juanito San Juanico, *sorry* where several gas sausages exploded.’

The sense of shame or embarrassment associated with this third type of situations is an experience regarded by psycholinguists as belonging to the class of "self-conscious emotions" and said to arise when individuals evaluate their actions in relation to some standard or rule of acceptable behavior and conclude that they have failed (Lewis et al. 1993). At the same time, this brings to mind the concept of "observed behavior (OB) face processes" recently formulated in Lacroix (2023), according to which speakers engage in self-oriented facework in situations where they think that the addressee, witnessing their improper behavior, will evaluate them negatively, and attempt to counter the potential negative judgement of their interlocutor in some way or another.

We believe that Lacroix's description fits our cases of no offense to the interlocutor. The self-conscious speaker, aware of his/her failure to comply with the rules of proper—or just adequately articulate—behavior, anticipates how this will affect his/her image in the eyes of the interlocutor and appeals to *perdón* to ward off the potential damage. It is a *perdón* stripped of its original illocutionary meaning ('I regret that I offended you'). The only message it carries is something along the lines of 'I know how I am expected to behave; my action should be seen as an unwonted slip'. In other words, the pragmatic inference, available but subordinated in genuine apologies ($perdón_1$), has been incorporated into the semantics of *perdón* and has given rise to a conventionalized or grammaticalized meaning which enables *perdón* to function as a self-face-saving device ($perdón_3$). Of course, $perdón_2$ and $perdón_3$ are proximate: they share the prominence of the self-oriented value but differ in that $perdón_2$ still contains a (weak) expression of regret for an acknowledged offense caused to the addressee.

From this point of view, the three meanings of *perdón* are susceptible to being ordered along a cline of decreasing intersubjectivity and increasing subjectivity. This goes against the traditional direction posited by Traugott (1999, p. 3) according to which intersubjectification follows, and arises from, subjectification, and which has been verified in a wide range of concrete cases to account for the evolution of DMs. However, this unidirectional shift from subjectivization to intersubjectivization has also been challenged in some studies, suggesting that the relation between the two notions should be thought as allowing for variable patterns of development (cf. among others Cornillie 2014; Hancil 2018 and references therein).

This in-depth analysis of *perdón* also enables us to tackle the question of what exactly differentiates $perdón_3$ from other DM like *bueno*, *oye/oiga*, *este*, etc. connected with similar metadiscursive functions (reformulations, hesitations, repetitions, etc.) and to pinpoint more precisely its specific contribution to this paradigm of DMs. Based on our empirical analysis, it is clear that $perdón_3$ addresses problems related to the flow of discourse in its own unique way, focusing the perspective on the image of self. The competing forms have different histories and introduce different nuances in the management of these conversational phenomena.

In a similar vein, $perdón_2$ invites an analysis in terms of an interactional type of connective marker (Brenes Peña 2021) or modal operator (Fuentes Rodríguez 2009) that, considering the functions it performs in contexts of minor offenses, is evaluated to be so slight that *perdón* is said to express a "pseudo-apology" (Brenes Peña 2021, pp. 156–57). In these cases, it is used to soften the impoliteness of interruptions, of requests for repetitions, of unwelcome responses to a petition, or of intrusions into one's territory with an attention-getting marker. Again, some of these functions have been attributed to other DMs (*bueno*, *oye/oiga*, *mira/mire*), and the relevant question hinges on what it is that $perdón_2$ accomplishes in these contexts, in contrast with the competing forms. We have grouped these contexts in our category of offensive behavior threatening the hearer's negative face, since they involve violations of one's claims to privacy and freedom from impositions and

impediments (in terms of Brown and Levinson 1987). We consider that, if speakers choose *perdón* over other markers in situations of this nature, it is because they feel that some words of apology are in order. However, sentiments of genuine regret and the hope of being forgiven are clearly absent. The main preoccupation concerns the negative impact the offense, however slight, will have on their public image. This balance between acknowledging the offense and attending interests of the self is precisely what *perdón*$_2$ helps to achieve, with its backgrounded apologizing value and its foregrounded message of face-redress ('I know the social rules and I usually respect them').

Interestingly, from a contrastive point of view, in our Mexican data, where *perdón* predominates, disagreements and criticisms, classified as more serious offenses on the grounds that they challenge the other's social dignity (his positive face-wants), still resist the use of *perdón* to some extent. We interpret this phenomenon as suggestive of the fact that the entrenchment of grammaticalized *perdón*$_3$, along with the expansion of the proximate *perdón*$_2$, have generated an implicit association of the form with issues of self-worth such that speakers hesitate to resort to the formula (*perdón*$_1$) in contexts where they evaluate their behavior as being truly offensive and choose more elaborate expressions of apology, such as *disculpa/e*, to convey their feeling of regret.

The Peninsular data, on the other hand, give evidence of a less advanced process of grammaticalization. The self-face-saving device (*perdón*$_3$) is frequent in contexts of no offense (to the interlocutor), but elsewhere, other IFIDs (especially *perdona/e*, *lo siento*), are still preferred. This suggests that, contrary to Peninsular Spanish, Mexico seems to have also regularized the use of *perdón* for offenses harming the negative face of the addressee, with a clear dominance in turn-taking contexts. This divergence between peninsular and Mexican *perdón* suggests a further stage of the latter in its grammaticalization towards discourse organization. This synchronic variation (Schneider and Barron 2008; Aijmer 2022) corroborates the well-known fact that the use of pragmatic markers not only differs between languages, but also between varieties of the same language and thus reveals diverging patterns of ongoing grammaticalization between both varieties.

## 5. Concluding Remarks

By means of a corpus-based comparative analysis, this study has examined the degree of grammaticalization of the apologetic marker *perdón* in Peninsular and Mexican Spanish, which has led to a number of significant insights situated at both the methodological and theoretical level of analysis.

First of all, from a methodological point of view, we approached the speech act of apology from the perspective of the two basic theoretical concepts inherent to an apology, namely the type of offense and the concept of face. The analytical tool used for this purpose is a taxonomy of offenses motivating apologies in Spanish, organized around the concept of face of both the speaker and the interlocutor. This taxonomy is shown to be a fruitful methodological tool for the analysis of apologetic markers that provides a systematic and verifiable alternative to more intuitive approaches. It does not only offer usage-based evidence for previous theorizing concerning the grammaticalization process of apologetic markers, but also leads to a gradual refinement of these previous results from a contrastive point of view.

Theoretically, the study offers a comprehensive perspective on the grammaticalization of *perdón* in Spanish. Contrary to what usually happens to other DMs, the point of departure for *perdón* rests in an element that already has an illocutionary force at the onset and this illocutionary material is recruited for further illocutionary and discursive uses. That is, the grammaticalization or discursisation process of *perdón* embodies a process that affects the illocutionary potential of the speech act itself. So, rather than semantic bleaching, this can be best described as a case of progressive weakening of its illocutionary force (Jucker 2019a). However, even in this grammaticalized use of the form, some important vestiges of the canonical model of apologies remain essential for the comprehension of its present-day uses: we have seen that the two theoretical notions underlying the canonical

definition of an apology—namely face and offense—are still determining the contemporary grammaticalized uses and values of *perdón*. At the same time, these vestiges also help to relate its contemporary grammaticalized values to the original illocutionary act of regret. More concretely, under our proposal, contemporary *perdón* is a polysemous item associated with three main pragmatic meanings. The interpretation of the formula depends on the concrete discourse contexts in which it is used and interacts in a critical way with the type of offense determining its appearance:

- *Perdón₁* is an expression of regret that is essentially hearer-supportive, but it comes along with a secondary and inferable message of interest to the speaker ('I know the norms; I usually do not offend people');
- *Perdón₂* relates to minor social infringements; the illocutionary component of regret is diluted, while the self-protective inferential message gains in prominence;
- *Perdón₃* is a grammaticalized self-face-saving device; the remedial move exclusively targets the face-wants of the seemingly apologizing individual.

That is to say, the bleaching of the original illocutionary force of the speech act of apology geared towards the interlocutor is compensated by a pragmatic strengthening of what used to be only a pragmatic inference with discursively secondary status in the case of genuine apologies. More precisely, the underlying meaning akin to 'I know the norms of appropriate behavior prevailing in our community and I usually act accordingly' has become conventionalized as a grammaticalized formulaic speech act. In this grammaticalized form, it is often used without the presence of a clear offense towards the interlocutor but converts into a self-face-saving device geared towards the speaker. In other words, the pragmatic inference, available but subordinated in genuine apologies (*perdon₁*), has been incorporated into the semantics of *perdón* and has given rise to a conventionalized or grammaticalized meaning which enables *perdón* to function as a self-face-saving device (*perdón₃*).

Interestingly, the degree of entrenchment of grammaticalized *perdón₃* seems to give rise to diverging patterns of grammaticalization across varieties of the same language. As such, we have seen that Mexican Spanish seems to have regularized the use of *perdón* not only as a self-face-saving device (*perdón₃*), but also frequently allows it for offenses harming the negative face of the addressee, with a clear dominance in turn-taking contexts. Peninsular Spanish, on the other hand, gives evidence of a less advanced process of grammaticalization. The self-face-saving device (*perdón₃*) is frequent in contexts of no offense (to the interlocutor), but, elsewhere, other IFIDs (especially *perdona/e*, *lo siento*) are still preferred. To conclude, the present study thus corroborates and at the same time refines Viberg's (1999) conclusion that grammaticalization can drive cognates apart semantically, as long as we interpret cognates both at the interlinguistic and intralinguistic/dialectic level.

**Author Contributions:** Conceptualization, M.J. and C.M.; Methodology, C.M. and J.E.A.B.; Investigation, M.J., C.M. and J.E.A.B.; Data curation, M.J. and J.E.A.B.; Writing—original draft, M.J. and C.M.; Writing—review & editing, M.J.; Visualization, M.J.; Supervision, M.J.; Project administration, M.J.; Funding acquisition, M.J. All authors have read and agreed to the published version of the manuscript.

**Funding:** This research was funded by Ghent University, grant number [BOF/STA/202009/031].

**Institutional Review Board Statement:** Not applicable.

**Informed Consent Statement:** Not applicable.

**Data Availability Statement:** The research data of this study is available in TROLLing. Jansegers, Marlies; Melis, Chantal; Arrington Báez, Jennie Elenor, 2023, Replication Data for: Diverging grammaticalization patterns across Spanish varieties: the case of *perdón* in Mexican and Peninsular Spanish, https://doi.org/10.18710/IEXVVN (accessed on 1 December 2023), DataverseNO.

**Conflicts of Interest:** The authors declare no conflict of interest.

## Notes

[1]    See Section 3.1 for more information related to the corpora used for this study.

[2]    Examples taken from González-Cruz (2012).

[3]    Although theoretically possible, our dataset does not contain any cases of offenses towards the speaker's negative face.

## References

### Corpus Source

*América y España español coloquial* [AMERESCO] [online]. https://esvaratenuacion.es/ (accessed on 1December 2023).

*Valencia Español Coloquial* [Val.Es.Co] [online]. https://www.valesco.es/ (accessed on 1 December 2023).

*Corpus del Habla de Baja California* [CHBC] [online]. http://www.corpus.unam.mx/geco/portal/index/chbc (accessed on 1 December 2023).

*Corpus Michoacano del Español* [CME]. Non published corpus, access courtesy of the authors.

*Corpus Oral de Madrid* [CORMA]. Non published corpus, access courtesy of the authors.

*Corpus Oral de Referencia del Español Contemporáneo* [CORLEC] [online]. http://www.lllf.uam.es/ESP/Corlec.html (accessed on 1 December 2023).

*Corpus Sociolingüístico de la Ciudad de México* [CSCM] [online]. https://lef.colmex.mx/corpus_sociolinguistico.html (accessed on 1 December 2023).

*Proyecto para el Estudio Sociolingüístico del Español de España y América* [PRESEEA] [online]. https://preseea.uah.es/ (accessed on 1 December 2023).

### Secondary Source

Aijmer, Karin. 1996. *Conversational Routines in English: Convention and Creativity*. Studies in Language and Linguistics. London: Longman.

Aijmer, Karin. 2022. "Well He's Sick Anyway Like": *Anyway* in Irish English. *Corpus Pragmatics* 6: 101125. [CrossRef]

Arnovick, Leslie K. 1999. *Diachronic Pragmatics: Seven Case Studies in English Illocutionary Development*. Amsterdam and Philadelphia: John Benjamins.

Bello, Bethany Ann. 2015. Atenuación e intensificación pragmáticas en la expresión de actividades de imagen: Un estudio contrastivo en conversaciones coloquiales del español y del inglés. Ph.D. thesis, University of Valencia, Valencia, Spain.

Blum-Kulka, Shoshana, Juliane House, and Gabriele Kasper. 1989. *Cross-Cultural Pragmatics: Requests and Apologies*. Norwood: Ablex.

Boye, Kasper. 2023. Grammaticalization as Conventionalization of Discursively Secondary Status: Deconstructing the Lexical-Grammatical Continuum. *Transactions of the Philological Society* 121: 270–92. [CrossRef]

Bravo, Diana. 1999. ¿Imagen positiva vs. imagen negativa? Pragmática socio-cultural y componentes de face. *Oralia* 2: 155–84.

Bravo, Diana. 2005. Categorías, Tipologías y Aplicaciones. Hacia una Redefinición de la Cortesía Comunicativa. In *Estudios de la (Des)Cortesía en Español. Categorías Conceptuales y Aplicaciones a Corpora Orales y Escritos*. Edited by Diana Bravo. Buenos Aires: Dunken, pp. 21–52.

Brenes Peña, Ester. 2021. De la petición de disculpas a la contraargumentación: Análisis macrosintáctico de perdón. In *Sintaxis Discursiva: Construcciones y Operadores en Español*. Edited by Catalina Fuentes Rodríguez, Ester Brenes Peña and Víctor Pérez Béjar. Bern: Peter Lang, pp. 135–71.

Brinton, Laurel J. 2017. *The Evolution of Pragmatic Markers in English: Pathways of Change*. Cambridge: Cambridge University Press. [CrossRef]

Brooks, Roy L. 1999. *When Sorry Isn't Enough: The Controversy over Apologies and Reparations for Human Injustice*. New York: New York University Press.

Brown, Penelope, and Stephen C. Levinson. 1987. *Politeness: Some Universals in Language Usage*. Cambridge: Cambridge University Press.

Castillo Lluch, Mónica. 2008. La formación de marcadores discursivos vaya, venga, anda y vamos. In *Actas del VII congreso internacional de historia de la lengua española*. Edited by Concepción Company Company. Madrid: Arco Libros, pp. 1739–52.

Chen, Rong. 2001. Self-Politeness: A Proposal. *Journal of Pragmatics* 33: 87–106. [CrossRef]

Claridge, Claudia, and Leslie Arnovick. 2010. Pragmaticalisation and Discursisation. In *Historical Pragmatics*. Edited by Andreas H. Jucker and Irma Taavitsainen. Berlin and New York: De Gruyter Mouton, pp. 165–92.

Cornillie, Bert. 2014. La historia de la complementación con parecer y resultar. Apuntes sobre la (inter)subjetivización. *Revista de la Sociedad Argentina de Lingüística* 1: 1–15.

Coulmas, Florian. 1981. Poison to Your Soul: Thanks and Apologies Contrastively Viewed. In *Conversational Routine*. Edited by Florian Coulmas. The Hague: Mouton, pp. 69–91.

Denoyelle, Corinne. 2020. La réalisation de l'excuse en moyen français: Une recherche en pragmatique historique. *Travaux de Linguistique* 2: 145–81. [CrossRef]

Detges, Ulrich, and Richard Waltereit. 2016. Grammaticalization and pragmaticalization. In *Manual of Grammatical Interfaces in Romance*. Edited by Susann Fischer and Christoph Gabriel. Berlin and Boston: De Gruyter, pp. 635–58.

Deutschmann, Mats. 2003. Apologizing in British English. Ph.D. thesis, University of Umeå, Umeå, Sweden.

Edmondson, Willis J. 1981. On Saying You're Sorry. In *Conversational Routine*. Edited by Florian Coulmas. The Hague: Mouton, pp. 273–88.

Fischer, Kerstin. 2006. *Approaches to Discourse Particles*. Amsterdam: Elsevier.

Fuentes Rodríguez, Catalina. 2009. *Diccionario de Conectores y Operadores del Español*. Madrid: Arco Libros.

Gancedo Ruiz, Marta. 2019. Evolución de la imagen del rol familiar en el teatro de finales del siglo XIX a mitad del XX. Su manifestación en la atenuación e intensificación de los actos directivos. Ph.D. thesis, University of Valencia, Valencia, Spain.

Ghezzi, Chiara, and Piera Molinelli. 2019. Italian *scusa* from politeness to mock politeness. *Journal of Pragmatics* 142: 245–57. [CrossRef]

Gibney, Mark, Rhonda Howard-Hassman, Jean-Marc Coicaud, and Nicklaus Steiner. 2007. *The Age of Apology: The West Faces Its Own Past*. Philadelphia: University of Pennsylvania Press.

Goffman, Erving. 1967. *Interaction Ritual: Essays on Face-to-Face Behavior*. New York: Doubleday-Anchor.

Goffman, Erving. 1971. *Relations in Public: Microstudies of the Public Order*. London: Allen Lane.

Goffman, Erving. 1972. On Face-Work: An Analysis of Ritual Elements in Social Interaction. In *Communication in Face-to-Face Interaction*. Edited by John Laver and Sandy Hutcheson. Harmondsworth: Penguin, pp. 179–96.

González-Cruz, María-Isabel. 2012. Apologizing in Spanish: A study of the strategies used by university students in Las Palmas de Gran Canaria. *Pragmatics* 22: 543–65. [CrossRef]

Hancil, Sylvie. 2018. (Inter)subjectification and paradigmaticization: The case study of the final particle but. In *New Trends in Grammaticalization and Language Change*. Edited by Sylvie Hancil, Tine Breban and José Vicente Lozano. Amsterdam: John Benjamins, pp. 291–313.

Harris, Sandra, Karen Grainger, and Louise Mullany. 2006. The Pragmatics of Political Apologies. *Discourse and Society* 17: 715–37. [CrossRef]

Haverkate, Henk. 1994. *La cortesía verbal: Estudio pragmalingüístico*. Madrid: Gredos.

Hernández Flores, Nieves. 2005. La cortesía como búsqueda del equilibrio de la imagen social: La oscilación de la imagen en un debate televisivo. In *Actas del II Coloquio del Programa EDICE*. Edited by Jorge Murillo Medrano. San José: Universidad de Costa Rica, pp. 37–52.

Hernández Flores, Nieves. 2008. Politeness and other types of facework: Communicative and social meaning in a television panel discussion. *Pragmatics* 18: 577–603. [CrossRef]

Holmes, Janet. 1990. Apologies in New Zealand English. *Language in Society* 19: 155–99. [CrossRef]

Holmes, Janet. 1998. Apologies in New Zealand English. In *The Sociolinguistics Reader: Gender and Discourse*. Edited by Jenny Cheshire and Peter Trudgill. London: Arnold, vol. 2, pp. 201–39.

Hopper, Paul J. 1991. On some principles of grammaticization. In *Approaches to Grammaticalization*. Edited by Elizabeth Closs Traugott and Bernd Heine. Amsterdam: John Benjamins, vol. 1, pp. 17–36.

Jucker, Andreas H. 2019a. Speech Act Attenuation in the History of English: The Case of Apologies. *Glossa: A Journal of General Linguistics* 4: 1–25. [CrossRef]

Jucker, Andreas H. 2019b. "Oops, I forgot, sorry": The spill cries oops and whoops in the history of American English. *Lingue e Linguaggi* 31: 15–33.

Kerbrat-Orecchioni, Catherine. 1992. *Les Interactions Verbales*. Paris: Armand Colin, vol. 2.

Kohnen, Thomas. 2017. Non-Canonical Speech Acts in the History of English. *Zeitschrift für Anglistik und Amerikanistik* 65: 303–18. [CrossRef]

Lacroix, René. 2023. Two phenomena behind the terminology of face. *Journal of Politeness Research* 19: 323–53. [CrossRef]

Lazare, Aaron. 2004. *On Apology*. Oxford: Oxford University Press.

Leech, Geoffrey Neil. 1983. *Principles of Pragmatics*. London: Longman.

Lehmann, Christian. 2015. *Thoughts on Grammaticalization*, 3rd ed. Berlin: Language Science Press. First published 1995.

Lewis, Michael, Jeannette M. Haviland-Jones, and Lisa Feldman Barrett, eds. 1993. *The Handbook of Emotion*. New York: Guilford Publications.

Márquez-Reiter, Rosina. 2000. *Linguistic Politeness in Britain and Uruguay: A Contrastive Study of Requests and Apologies*. Amsterdam: John Benjamins.

Medina López, Javier. 2023. Formas de perdón, arrepentimiento y disculpas en la historia del español: Una aproximación desde el análisis textual. *Nueva Revista de Filología Hispánica* 71: 499–529. [CrossRef]

Meier, Ardith J. 1995. Passages of politeness. *Journal of Pragmatics* 24: 381–92. [CrossRef]

Molina, Clara. 2011. Routes for development in the pragmaticalization of *sorry* as a formulaic marker. *Revista Alicantina de Estudios Ingleses* 24: 191–212. [CrossRef]

Norrick, Neal R. 1978. Expressive Illocutionary Acts. *Journal of Pragmatics* 2: 277–91. [CrossRef]

Norrick, Neal R. 1979. The lexicalization of pragmatic functions. *Linguistics* 17: 671–86. [CrossRef]

Ogiermann, Eva. 2006. Cultural variability within Brown and Levinson´s politeness theory. English, Polish and Russian apologies. In *Studies in Contrastive Linguistics*. Edited by Cristina Mourón-Figueroa and Teresa Moralejo-Gárate. Santiago de Compostela: Universidade de Santiago de Compostela Publicacións, pp. 707–18.

Ogiermann, Eva. 2009. *On Apologising in Negative and Positive Politeness Cultures*. Amsterdam and Philadelphia: John Benjamins.

Ogiermann, Eva. 2015. Apology discourse. In *The International Encyclopedia of Language and Social Interaction*. Edited by Karen Tracy. Chichester: John Wiley & Sons, pp. 1–16. [CrossRef]

Olshtain, Elite. 1989. Apologies Across Languages. In *Cross-Cultural Pragmatics: Requests and Apologies*. Edited by Shoshana Blum-Kulka, Juliane House and Gabriele Kasper. Cambridge: Cambridge University Press, pp. 155–73.

Olshtain, Elite, and Andrew Cohen. 1983. Apology: A Speech Act Set. In *Sociolinguistics and Language Acquisition*. Edited by Nessa Wolfson and Elliot Judd. Rowley: Newbury House, pp. 18–36.

Owen, Marion. 1983. *Apologies and Remedial Interchanges. A Study of Language Use in Social Interaction*. Berlin: Mouton.

Schneider, Klaus Peter, and Anne Barron, eds. 2008. *Variational Pragmatics. A Focus on Regional Varieties in Pluricentric Languages*. Amsterdam: John Benjamins.

Searle, John R. 1969. *Speech Acts*. Cambridge: Cambridge University Press.

Traugott, Elizabeth C. 1999. From subjectification to intersubjectification. Paper presented at the Workshop of Historical Pragmatics, Fourteenth International Conference on Historical Linguistics, Vancouver, BC, Canada, August 9–13.

Traugott, Elizabeth C. 2003. Constructions in grammaticalization. In *The Handbook of Historical Linguistics*. Edited by Brian D. Joseph and Richard D. Janda. Oxford: Blackwell, pp. 624–47.

Trosborg, Anna. 1987. Apology strategies in natives/nonnatives. *Journal of Pragmatics* 11: 147–67. [CrossRef]

Trosborg, Anna. 1995. *Interlanguage Pragmatics: Requests, Complaints and Apologies*. Berlin and New York: De Gruyter Mouton.

Van Bogaert, Julie. 2011. "I Think" and Other Complement-Taking Mental Predicates: A Case of and for Constructional Grammatical-ization. *Linguistics* 49: 295–332. [CrossRef]

Viberg, Ake. 1999. The polysemous cognates Swedish *gå* and English *go*: Universal and language-specific characteristics. *Languages in Contrast* 2: 87–113. [CrossRef]

Wagner, Lisa C. 1999. Towards a Sociopragmatic Characterization of Apologies in Mexican Spanish. Ph.D. thesis, The Ohio State University, Columbus, OH, USA.

Wagner, Lisa. C. 2004. Positive- and negative-politeness strategies: Apologizing in the speech community of Cuernavaca, Mexico. *International Communication Studies* 13: 19–27.

Wichmann, Anne, Anne-Marie Simon-Vandenbergen, and Karin Aijmer. 2010. How prosody reflects semantic change: A synchronic case study of of course. In *Subjectification, Intersubjectification and Grammaticalization*. Edited by Kristin Davidse, Lieven Vandelanotte and Hubert Cuyckens. Berlin and New York: De Gruyter Mouton, pp. 103–54.

Williams, Graham. 2018. *Sincerity in Medieval English Language and Literature*. London: Palgrave Macmillan.

Winter-Froemel, Esme. 2014. Re(de)fining grammaticalization from a usage-based perspective Discursive ambiguity in innovation scenarios. *Folia Linguistica* 48: 503–56. [CrossRef]

Zipf, George Kingsley. 1949. *Human Behavior and the Principle of Least Effort: An Introduction to Human Ecology*. New York: Hafner Publishing Company.

