# Peer review of "Diverging Grammaticalization Patterns across Spanish Varieties: The Case of perdón in Mexican and Peninsular Spanish"

_languages, doi:10.3390/languages9010013_

Round 1

Reviewer 1 Report

Comments and Suggestions for Authors

Various corrections need to be made. They are included in the attached file.

Author Response

We would like to thank the reviewer for his/her valuable and very detailed comments and suggestions. We have revised our paper as advised by the referee and hope to have come to a publishable version. Please find attached our response to the reviewer. Our comments (in blue) are inserted directly into the review.

Reviewer 2 Report

Comments and Suggestions for Authors

See my comments and suggestions in the attached PDF file. 

Comments on the Quality of English Language

I have one minor translation recommendation for a data point.  See the attached file with specific comments. 

Author Response

(The authors gave the same response as above.)

Reviewer 3 Report

Comments and Suggestions for Authors

Interesting and well-written paper that reflects on the state of grammaticalisation of the marker “perdón” in two varieties of Spanish (Mexican and Peninsular). Although the work is very well documented and presents interesting conclusions (based on a classification of the offending act which allows the author to establish the differences in the use of “perdón” in the two varieties), we believe that, on occasions, this description takes second place, focusing more on theoretical aspects of the processes of grammaticalisation. Thus, the paper lacks the analysis of more contextualised examples of “perdón”. In fact, there is hardly any reference to the examples analysed in the first pages, not even to exemplify the 3 types of usage described (line 640). It would also be appreciated to include more examples from line 659 onwards.

Comments:

- The introduction is too long and does not establish from the beginning what the hypotheses to be studied are. The theoretical part is developed a lot (in a very precise way), but it would be convenient to explain what the study consists of at the empirical level.

- From line 178, the author's position on grammaticalisation - pragmaticalisation could be explained, although the question is developed later. In line 182, the author could announce whether he/she are going to try to answer the questions raised.

- Review the use of double and single inverted commas. Sometimes single quotation marks are used when doble ones should appear.

- In line 314 the author could announce why the results of the theoretical studies are contrary to the empirical results obtained.

- Line 333: Sometimes they are not synonyms, for example in (5) and (6).

- In line 339, explain how the 500 “perdón” occurrences of the analysed sub-corpus have been selected.

- Table 1: explain whether the taxonomy follows an order, in each column, from "most offensive" to "least offensive", or if there is no order at all.

- Line 390: avoid the verb "discover" and replace it with "define" or "describe".

- From line 536: instead of comparing the use of new markers (i.e., bueno, este), it would be more pertinent to go back to the next terms and expressions presented above (lo siento...). Add examples to illustrate on these lines.

- Line 646: add that, at this degree of grammaticalisation, “perdón” also is employed to organise and reformulate the discourse (as is clear from the examples).

Other minor corrections are suggested in the attached document.

Author Response

(The authors gave the same response as above.)

Reviewer 4 Report

Comments and Suggestions for Authors

This is an exciting paper on the grammaticalization of perdón in two varieties of Spanish. The author shows an understanding of the issue at hand, particularly in the literature review. The findings are important for the field of Pragmatics in general and for the theorization on Grammaticalization and studies in Spanish variation in particular. However, I had some issues adequately following the section of the discussion. Besides, some stylistic improvements must be implemented before publishing this paper. I am sure that with some careful attention paid to the writing style and to ordering some ideas, this paper will become suitable for publication.

Perhaps section 4. Discussion: diverging grammaticalization patterns should be divided into subsections (grammaticalization, interpersonal implications, meanings of perdón…), or the train of thought should be more precise. Following the author’s reasoning for relating the topics between paragraphs is sometimes complicated as the text develops. Please review this section and make it more coherent for the reader.

Some minor grammatical mistakes were fixed in the revision of the paper. I may have missed some, so I encourage the author to read over the paper with this in mind.

Below is a list of specific suggestions. Some things I have marked once or twice on the manuscript, but the author should check for more occurrences throughout the paper. Please check the attachment for more comments and suggestions.

·      Reduce the length of some sentences. I have given some suggestions (page 5, lines 240-243; page 6, line 289…), but these are not exhaustive. I have underlined some sentences that I think should be cut shorter, but a careful review should be done.

·      I would recommend repeating the name of the researcher cited instead of using pronouns (‘he’, ‘she’), as it allows the reader to identify the source more quickly and is a form of ‘respect’ toward the other author(s).

·      Some words are redundant (‘indeed’ line 265, ‘that is’ line 278…). I have indicated these suggestions by crossing the item(s) in question.

·       ‘Spoken’ is preferred for corpus instead of ‘oral’, so please apply these changes in the paper and the abstract.

·      In English academic writing, the first-person singular ‘I’, ‘me’, ‘my’ is preferred instead of royal ‘we’, ‘us’, ‘our’. I advise using the passive voice if the author feels contradicted by the first-person singular. Please disregard this comment if there are multiple authors to this paper. Some examples are:

o   ‘This first step will allow us to quantitatively assess the overall frequency and productivity of perdón compared to its near-synonymous expressions” (page 7, line 334) > ‘This first step will allow a quantitative assessment of the overall…’

o   ‘As we saw in the taxonomy of offenses presented in Table 1’ > ‘As seen in ((the taxonomy of offenses presented)) in Table’ (page 9)

·      The tense used in the paper should be consistent. If the author talks about the analyses in the future tense (i.e., ‘this step will allow us to…’), it should be used throughout the paper (primarily the methodology and conclusions). I would advise the present tense as a description, or the past, retelling what has already been done in this piece of research.

·      a note about the transcription system should be made in the methodology.

·      Please review the examples, particularly the translation and the transcription system. I would recommend including not only the corpus but also the name of the particular dataset (conversation, interview)

·      Other suggestions were made throughout the paper (l. 115, l.124, l. 340…), so please check all comments.

Comments on the Quality of English Language

My comments on the quality of English are included in the general suggestions section.

Author Response

(The authors gave the same response as above.)

Round 2

Reviewer 4 Report

Comments and Suggestions for Authors

Thank you for applying the changes I suggested, as well as the other reviewer's. I now think this paper is suitable for publication. Great work!